# Preparation of Magnetic MIL-68(Ga) Metal–Organic Framework and Heavy Metal Ion Removal Application

**DOI:** 10.3390/molecules27113443

**Published:** 2022-05-26

**Authors:** Youjun Zhang, Licong Liu, Dixiong Yu, Jinglan Liu, Lin Zhao, Jinpeng Liu, Shuo Liu

**Affiliations:** 1School of Environmental Science and Engineering, Tianjin University, Tianjin 300350, China; zhangyoujun001@126.com; 2Tianjin North China Geological Exploration Bureau, Tianjin 300170, China; liujinglanok@126.com (J.L.); zhaolin@163.com (L.Z.); 3Technical Centre of Solid Waste and Soil Ecology of Inner Mongolia, Huhehaote 010000, China; liulicong@163.com; 4Key Laboratory of Molecular Microbiology and Technology, Ministry of Education, Department of Microbiology, College of Life Sciences, Nankai University, Tianjin 300071, China; nkuydx@163.com; 5Tianjin Key Laboratory of Environmental Remediation and Pollution Control, College of Environmental Science and Engineering, Nankai University, Tianjin 300350, China; saub1986@126.com

**Keywords:** MOF, magnetic separation, heavy metal ions, adsorption

## Abstract

A magnetic metal–organic framework nanocomposite (magnetic MIL-68(Ga)) was synthesized through a “one pot” reaction and used for heavy metal ion removal. The morphology and elemental properties of the nanocomposite were characterized by scanning electron microscopy (SEM), Fourier transform infrared (FT-IR), X-ray powder diffraction (XRD), as well as zeta potential. Moreover, the factors affecting the adsorption capacity of the nanocomposite, including time, pH, metal ion type and concentration, were studied. It was found that the adsorption capacity of magnetic MIL-68(Ga) for Pb^2+^ and Cu^2+^ was 220 and 130 mg/g, respectively. Notably, the magnetic adsorbents could be separated easily using an external magnetic field, regenerated by ethylenediaminetetraacetic acid disodium salt (EDTA-Na_2_) and reused three times, in favor of practical application. This study provides a reference for the rapid separation and purification of heavy metal ions from wastewater.

## 1. Introduction

With the development of economy and society, heavy metal pollution in surface water and groundwater has become a great threat to the ecological environment and human health. Generally, heavy metal ions are not biodegraded, interact strongly with proteins in organisms, and are highly harmful to living beings, causing both acute and chronic poisoning [1,2,3,4,5]. Therefore, the removal of toxic heavy metal ions from water bodies, especially in wastewater, is of great importance.

So far, precipitation [6,7], membrane process [8,9], ion exchange [10,11] and adsorption [12,13] techniques have been demonstrated to effectively eliminate heavy metal ions from contaminated water. Among them, adsorption is a promising and effective method for practical use due to its high efficiency, simple design and easy operation [14,15,16,17]. Various adsorbents, such as carbon materials (activated carbon, nanotubes), polymers, metallic and metal compounds (nanoparticles, metal–organic frameworks, and magnetic materials), and minerals (silica, zeolites, and clays) have been developed to remove heavy metal ions in water [18,19,20,21]. As a kind of organic–inorganic hybrid nanomaterial, metal–organic frameworks (MOFs) possess the advantages of high surface area, adjustable porosity, better functionality, and easy design and synthesis, making them highly desirable in various research fields, such as adsorption [22,23,24,25,26]. For example, Zhang et al. found that the carboxyl group of MIL-based MOFs (MIL-121) is vital for selective adsorption towards Cu^2+^ [27]. However, the separation of MOFs after the adsorption process is laborious due to the need for centrifugation or filtration. Due to the magnetism, stability, and cost effectiveness, magnetic nanoparticles (MNPs) have been introduced in the preparation of MOFs. The obtained magnetic MOFs can easily be separated after the adsorption process with the aid of an external magnetic field and the post-processing is significantly simplified [28,29]. For example, Mahmoodi et al. synthesized a magnetic eggshell membrane-zeolitic imidazolate framework. The nanocomposite exhibited better adsorption performance for Cu^2+^ and dye than untreated zeolitic imidazolate framework [30]. Nonetheless, the development of novel magnetic adsorbents with simplicity and effectiveness for heavy metal ions removal from water is still urgent.

Materials of Institute Lavoisier frameworks (MILs) are a kind of MOF with good water stability, making them promising adsorbents for heavy metal ions in water [31,32]. Moreover, the central metal of MIL-68, Ga^3+^ has been demonstrated to play a key role in the biomedical field, such as anticancer, antibacteria, and hemostasis [33,34]. Until now, there have been few studies on gallium-based MOFs as adsorbents for heavy metal ion removal [35]. In this study, we designed and prepared a magnetic MIL-68 (Ga) MOF as an adsorbent in heavy metal ion removal for the first time via a one-pot reaction. XRD, SEM, FT-IR, zeta potential, and VSM were used to characterize the nanocomposite. The synthesized magnetic MOF exhibited effective adsorption capacity for Pb^2+^ and Cu^2+^ in aqueous solution, while it could be easily separated using an external magnetic field. The influence of pH, incubation time, and initial concentration of metal ions were investigated.

## 2. Results and Discussion

### 2.1. Characterization of MIL-68(Ga) and Magnetic MIL-68(Ga)

MIL-68(Ga) and magnetic MIL-68(Ga) were prepared via the solvothermal method with slight modification [36]. The morphology of magnetic MIL-68(Ga) was characterized by SEM. As shown in Figure 1a, MIL-68(Ga) appeared as rod-like crystallites of 1–2 µm length, which was a little shorter than the reported result [37]. For magnetic MIL-68(Ga), MIL-68(Ga) grew homogeneously around the Fe_3_O_4_ nanoparticles (Figure 1b).

EDS mapping experiments were further performed to analyze the elemental composition and distribution. As shown in Figure 2a, MIL-68(Ga) exhibited uniform distributions of the C, Ga, and O elements. Figure 2b shows the distribution of C, Ga, O, and Fe elements in magnetic MIL-68(Ga). Obviously, the Fe element uniformly distributed in the nanocomposites, suggesting the existence of Fe_3_O_4_ in MIL-68(Ga).

Appendix A illustrates the FT-IR spectra of MIL-68(Ga) and magnetic MIL-68(Ga). Both of the composites had obvious adsorption peaks in a wavelength range of 1700–1400 cm^−1^, demonstrating the existence of carbonyl groups [38]. Moreover, the new band at 580 cm^−1^ represented the presence of an Fe-O group in magnetic MIL-68(Ga).

The magnetic properties of magnetic MIL-68(Ga) were also characterized by VSM. As shown in Figure 3a, the saturation magnetization of Fe_3_O_4_ and magnetic MIL-68(Ga) was 33, and 12 emu g^−1^, respectively. It reveals clearly that the magnetic MIL-68(Ga) could be easily separated from water with a common magnet within 1 min (Figure 3b). Therefore, magnetic MIL-68(Ga) could be easily collected with an external magnetic field, which was appropriate in practical use.

The crystalline phase structure of MIL-68(Ga) and magnetic MIL-68(Ga) was also examined using XRD. As shown in Figure 4, the diffraction peaks of MIL-68(Ga) and magnetic MIL-68(Ga) in the range of 10 to 20 degrees were in good agreement with the characteristic peaks of the reported MIL-68(Ga) [36,37], while the peaks at 30–70 degrees were consistent with Fe_3_O_4_ [39]. Additionally, the surface area of magnetic MIL-68(Ga) was 865 m^2^/g, calculated with the BET model.

### 2.2. Adsorption Studies

#### 2.2.1. The Adsorption Performance of Magnetic MIL-68(Ga) in Removing Heavy Metal Ions

The adsorption performance of magnetic MIL-68(Ga) on common heavy metal ions, such as Pb^2+^, Cu^2+^, and Cd^2+^, was first evaluated by measuring the remaining metal ions in the solution after adsorption. Magnetic MIL-68(Ga) (200 mg/L) was incubated in 1 mL aqueous solution (containing 50 mg/L of metal ions, pH = 7) at room temperature for 30 min. Then the solution was separated using a magnetic field, and the concentration of metal ions in the solution was quantified by ICP-MS. Based on the adsorption capacity shown in Figure 5, though magnetic MIL-68(Ga) exhibited weak adsorption capacity towards Cd^2+^, it could remove Pb^2+^ and Cu^2+^ from the solution efficiently, which was mainly ascribed to its carbonyl groups [35].

#### 2.2.2. Effect of Contact Time on Pb^2+^ Adsorption

The effect of contact time on adsorption capacity of Pb^2+^ was then investigated. As shown in Figure 6, the magnetic MIL-68(Ga) exhibited obvious Pb^2+^ adsorption capacity, and the adsorption efficiency increased rapidly upon increasing the time to 30 min, and then nearly remained the same. Consequently, the optimal contact time was approximately 30 min.

#### 2.2.3. Effect of pH on Pb^2+^ Adsorption and Stability of Magnetic MIL-68(Ga)

pH is an important parameter to be considered in adsorption studies as pH alteration might change the surface charge and stability of the adsorbent. To test the effect of pH, the removal efficiency of Pb^2+^ was investigated in a pH range of 3 to 7. As shown in Figure 7a, upon increasing the pH from 3 to 7, the removal capacity of Pb^2+^ increased from 20 to 140 mg/g.

Moreover, we found that magnetic MIL-68(Ga) was sensitive to pH. As shown in Figure 7b, after incubation for 30 min, more than 50% of magnetic MIL-68(Ga) was broken down when the pH below 5, while it was stable at pH 7. Based on the above results, the optimum pH was obtained at 7. The SEM and EDS mapping spectra of magnetic MIL-68(Ga) after incubation at pH 3 also showed that the structure of MIL-68(Ga) collapsed while the magnetic Fe_3_O_4_ core was stable (Appendix A).

The lower adsorption capacity at an acidic pH could also be explained by the zeta potential value of adsorbents. As shown in Appendix A, the zeta potential value of magnetic MIL-68(Ga) increased significantly, from 9.2 mV at pH = 7 to 49 mV at pH = 3. As a result, the adsorption capacity decreased due to the electrostatic repulsion. Consequently, the carboxyl groups in magnetic MIL-68(Ga) were mainly responsible for metal ion adsorption.

#### 2.2.4. Effect of Initial Concentration on Pb^2+^ and Cu^2+^ Adsorption

With the optimal pH and incubation time, the adsorption capacity of magnetic MIL-68(Ga) on Pb^2+^ and Cu^2+^ was examined in a range of 10–200 mg/L. As shown in Figure 8, by increasing the initial concentration of metal ions, the removal capacity increased from about 20 to more than 200 mg/g for Pb^2+^, and from about 20 to 130 mg/g for Cu^2+^, suggesting better adsorption capacity for Pb^2+^. As a control, the adsorption capacity of MIL-68(Ga) for Pb^2+^ and Cu^2+^ was also examined. As shown in Appendix A, the Cu^2+^ adsorption capacity of MIL-68(Ga) was the same as magnetic MIL-68(Ga). Excitingly, the Pb^2+^ adsorption capacity of magnetic MIL-68(Ga) was much higher than MIL-68(Ga). Considering the simpler separation method, magnetic MIL-68(Ga) is much more effective than MIL-68(Ga) in the application of heavy metal ion removal.

To further investigate the adsorption capacity of magnetic MIL-68(Ga) for heavy metal ions in water, the effect of co-existing ions on the adsorption of the Pb^2+^ and Cu^2+^ was studied. As shown in Appendix A, the Pb^2+^ adsorption capacity of magnetic MIL-68(Ga) was stronger than Cu^2+^ (162 and 101 mg/g, respectively), and the two ions exhibited a competition relationship as the adsorption capacity for both ions was lower than only one ion (210 and 130 mg/g, respectively).

#### 2.2.5. Reusability of Magnetic MIL-68(Ga)

Reusability of the adsorbent is one of the important parameters for practical applications [30]. After adsorption of Pb^2+^, the magnetic MIL-68(Ga) was magnetically separated and regenerated by desorbing metal ions using EDTA-Na_2_ (1:1 to the molar of metal ions) [40]. As shown in Appendix A, the adsorption capacity of the regenerated adsorbent was 171 mg/g after three cycles, suggesting good reusability of magnetic MIL-68(Ga). Taken together, the magnetic MIL-68(Ga) exhibited high adsorption capacity for Pb^2+^ and Cu^2+^ at neutral pH and room temperature. Considering their simple preparation, easy regeneration and reusability, the composites can be used to remove heavy metal ions in water.

## 3. Materials and Methods

### 3.1. Materials

Iron(III) chloride hexahydrate, sodium acetate, ethylene glycol, gallium trinitrate, 1,4-dicarboxybenzene (BDC), DMF, cupric nitrate trihydrate, lead nitrate, and cadmium nitrate tetrahydrate (all analytical reagent grade) were purchased from Shanghai Aladdin Biochemical Technology Co., Ltd. (Shanghai, China). Deionized (D.I.) water was used to prepare all solutions.

### 3.2. Methods

#### 3.2.1. Preparation of Fe_3_O_4_ Particles

The magnetic particles were prepared by means of a solvothermal reaction [41]. Briefly, 2.70 g of FeCl_3_•6H_2_O and 7.20 g of sodium acetate were dissolved in 100 mL of ethylene glycol under vigorous stirring. The resulting homogeneous yellow solution was transferred to a Teflon-lined stainless-steel autoclave, sealed, and heated at 200 °C. After the reaction was allowed to proceed for 8 h, the autoclave was cooled to room temperature. The resulting black magnetite particles were washed several times with ethanol and dried in vacuum at 60 °C for 12 h. The average size of magnetite particles was about 300 nm.

#### 3.2.2. Preparation of MIL-68(Ga) and Magnetic MIL-68(Ga)

The gallium-based MOF, MIL-68(Ga), was synthesized from a mixture of gallium nitrate (0.512 g) and BDC (0.4 g) in 20 mL DMF [36]. The reactants were magnetically stirred and heated for 10 h at 100 °C. The mixture was centrifuged (8000 rpm, 5 min) and the obtained white precipitate was washed with DMF three times.

For the preparation of magnetic MIL-68(Ga), Fe_3_O_4_ particles (20 mg), gallium nitrate (0.512 g), and BDC (0.4 g) were mixed in 20 mL DMF. The reactants were ultrasonically dispersed and then magnetically stirred at 100 °C for 10 h. The mixture was centrifuged (8000 rpm, 5 min) and the precipitate was washed with DMF three times.

### 3.3. Characterization

The morphology and energy dispersive spectroscopy (EDS) mapping of magnetic MIL-68(Ga) were performed by scanning electron microscope (SEM) (TESCAN MIRA LMS, Brno, Czech). The crystallinity of the synthesized MIL-68(Ga) powder and magnetic MIL-68(Ga) composite was determined using an X-ray diffractometer (XRD) (Bruker D2 PHASER, Karlsruhe, Germany). Inductively Coupled Plasma Mass Spectrometry (ICPMS) (Agilent 7700, Palo Alto, Santa Clara, CA, USA) was used to measure the concentration of Pb^2+^, Ga^2+^ and Cu^2+^. Magnetic properties of the materials were characterized using a vibrating sample magnetometer (VSM) (LakeShore7404, Columbus, OH, USA). The functional groups of magnetic MIL-68(Ga) were detected by a Fourier transform infrared (FT-IR) spectrometer (Thermofisher Scientific, Nicolet iS50, Waltham, MA, USA). Zeta potential was analyzed by Nano-particle analyzer (Malvern Panalytical, Zetasizer Nano ZS, Malvern, UK). Brunauer–Emmett–Teller (BET) surface area analysis was performed by surface area and pore analyzer (Micromeritics, ASAP 2460, Norcross, GA, USA).

### 3.4. Adsorption Experiment

D.I. water, Pb(NO_3_)_2_ and Cu(NO_3_)_2_•3H_2_O were used to prepare the stock solution of Pb^2+^ and Cu^2+^ (2000 mg/L). Other Pb^2+^ or Cu^2+^ solutions with different concentrations were freshly prepared by gradient dilution of the stock solution.

To evaluate the MIL-68(Ga) and magnetic MIL-68(Ga) adsorption capacities for heavy metal ions, 200 mg/mL of MIL-68(Ga) or magnetic MIL-68(Ga) was mixed with a series concentration of Pb^2+^ and Cu^2+^ (10, 50, 100 and 200 mg/L) with different time and pH. Then the mixture containing MIL-68(Ga) was centrifuged, while the mixture containing magnetic MIL-68(Ga) was magnetically separated to obtain the liquid supernatant. ICP-MS was employed to determine the remaining metal ions in the solution.

The adsorption capacity of metal ions (C, mg/g) was calculated by (initial concentration of metal ions − equilibrium concentration of metal ions) × the volume of solution/the amount of adsorbents.

## 4. Conclusions

This study demonstrated a simple and efficient method for the synthesis of magnetic MIL-68(Ga), which was further characterized and identified by XRD, VSM, SEM and EDS mapping. The magnetic MIL-68 exhibited good adsorption capacity for heavy metal ions (Pb^2+^ and Cu^2+^) with an easy separation process using an external magnetic field. In summary, the magnetic MIL-68(Ga) is expected to be a promising candidate for heavy metal ion removal from practical aqueous solution.

## Figures and Tables

**Figure 1 molecules-27-03443-f001:**
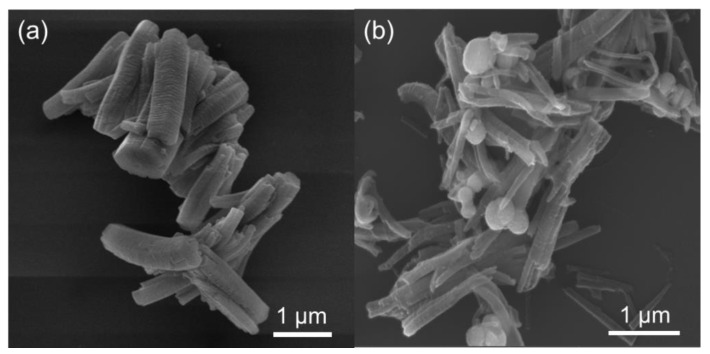
SEM images of MIL-68(Ga) (**a**) and magnetic MIL-68(Ga) (**b**).

**Figure 2 molecules-27-03443-f002:**
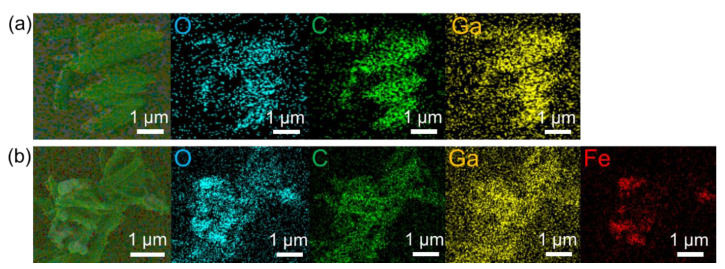
EDS mapping images of MIL-68(Ga) (**a**) and magnetic MIL-68(Ga) (**b**).

**Figure 3 molecules-27-03443-f003:**
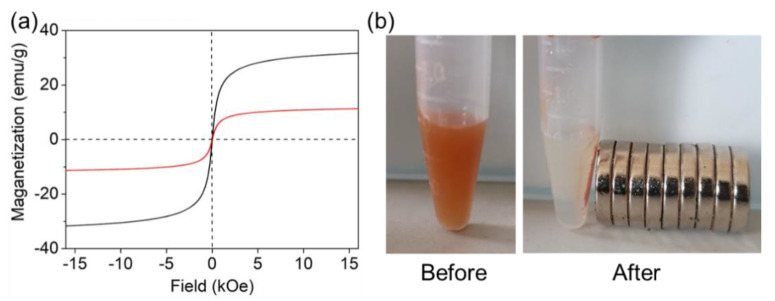
VSM curve of magnetic MIL-68(Ga) (**a**) and photographs of the magnetic MIL-68(Ga) solution before and after magnetic separation (**b**).

**Figure 4 molecules-27-03443-f004:**
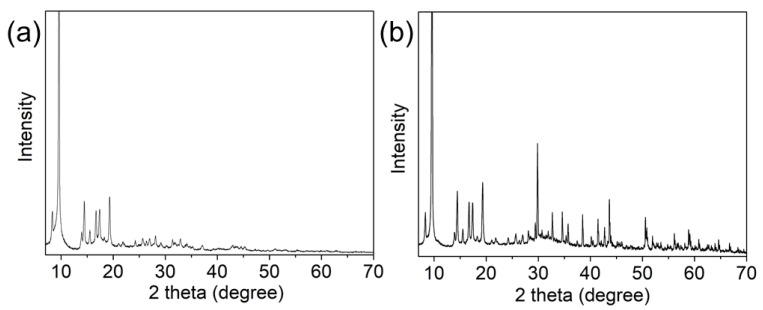
XRD pattern of MIL-68(Ga) (**a**) and magnetic MIL-68(Ga) (**b**).

**Figure 5 molecules-27-03443-f005:**
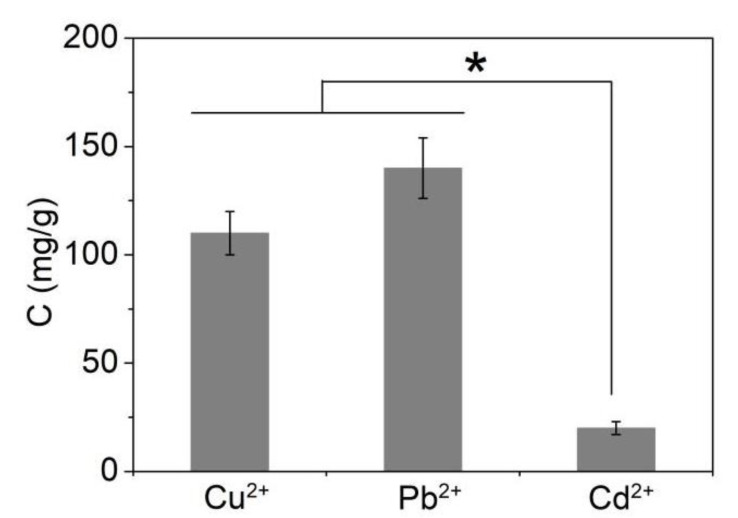
The adsorption capacity of magnetic MIL-68(Ga) in the removal of different heavy metal ions (50 mg/L of metal ions, 200 mg/mL of adsorbent, 1 mL solution, pH = 7, room temperature, 30 min of incubation time). The asterisk (*) indicates significant difference between the groups (*p* < 0.05).

**Figure 6 molecules-27-03443-f006:**
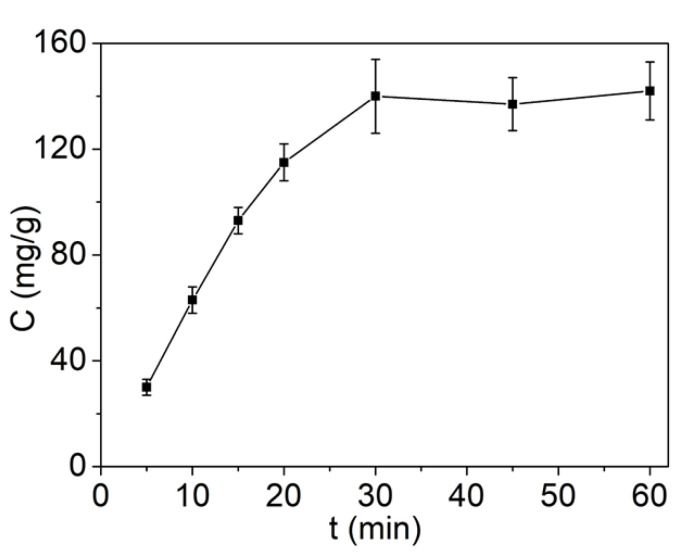
Effect of contact time on Pb^2+^ adsorption (50 mg/L of Pb^2+^, 200 mg/mL of adsorbent, 1 mL solution, pH = 7, room temperature).

**Figure 7 molecules-27-03443-f007:**
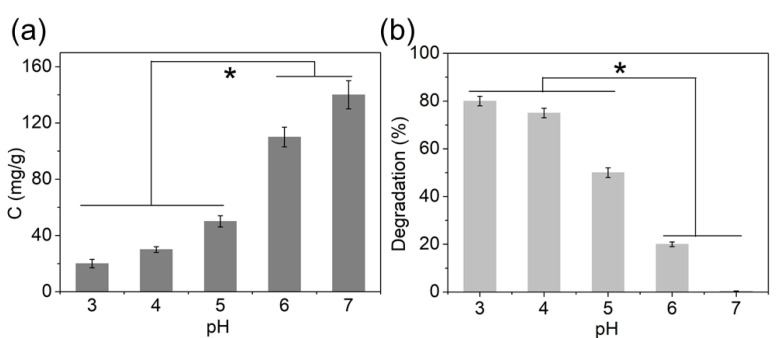
Effect of pH on (**a**) Pb^2+^ adsorption capacity (50 mg/L of Pb^2+^, 200 mg/mL of adsorbent, 1 mL solution, room temperature, 30 min of incubation time) and (**b**) the stability of magnetic MIL-68(Ga) (200 mg/mL of adsorbent, 1 mL solution, room temperature, 30 min of incubation time). The asterisks (*) indicate significant difference between the groups (*p* < 0.05).

**Figure 8 molecules-27-03443-f008:**
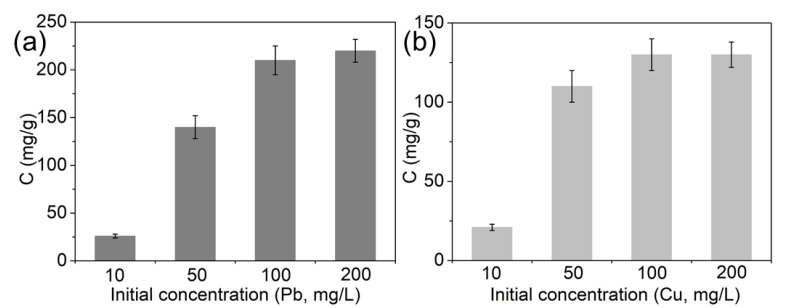
Adsorption capacity of magnetic MIL-68(Ga) on Pb^2+^ (**a**) and Cu^2+^ (**b**). (50 mg/L of metal ions, 200 mg/mL of adsorbent, 1 mL solution, pH = 7, room temperature, 30 min of incubation time).

## Data Availability

Not applicable.

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
