# Peer review of "Preparation of Magnetic MIL-68(Ga) Metal–Organic Framework and Heavy Metal Ion Removal Application"

_molecules, 2022, doi:10.3390/molecules27113443_

Round 1

Reviewer 1 Report

This manuscript shows some interesting results on adsorptive removal of metal ions. However, there are some revisions for publishing it on the Journal “molecules”.

Abstract: description on experimental results obtained in this study should be added.

Line 29: “removal of the toxic heavy metal ions” is changed to “removal of toxic heavy metal ions”.

Line 41: “et al” is changed to “et al.”.

Line 42: The term “for selectively adsorb of” is a grammatical error.

Line 49: The term “for heavy metal ions removal in water” is a grammatical error.

Line 58: Gallium is changed to gallium.

Line 101: “the” before magnetic MIL-68(Ga) should be deleted.

Line 139: I can’t understand the term “experiments contained 200 mg/L”.

Line 145: Can the author(s) show the experimental or instrumental evidence of the presence of carbonyl groups?

Line 148: Caption of Figure 5, the term “The asterisks indicate” is changed to “The asterisk indicates".

Line 150-151: This sentence should be changed.

Line 159-162: The mechanism of adsorption of metal ions is required.

Line 178: I can’t understand the value of “-20”.

Reviewer 3 Report

This manuscript is missing few important points that must be addressed before it can be considered for publication:
1. A precise, detailed discussion as to why the reported materials are important to be tested for water treatment or otherwise valuable to the environmental community. Simply saying that application of these materials as alternative for wastewater treatment has gained increased attention is not acceptable.
2. A clear, detailed description of how the materials were fabricated and the exposed areas/amounts could be considered as real for water treatment.
3. A strong justification in favor of the metals treatment by the proposed technology, explaining why the experimental conditions, including the materials used, is important.
4. A clear explanation of the chemistry/physics that describe the elimination of metals from solutions. A general description is not novelty. 

5. What is the real area of the material?

6. Scale in mapping imagens is needed.

7. reuse and recovery should be discussed.

8. lixiviation should be tested after the elimination of metals.

9. Experimental info should be reported in the figure captions.  

Round 2

Reviewer 1 Report

The authors carefully and properly modified the document.

I have some requests before publication. 

Abstract: More experimental results are added for other researchers or readers to understand this investigation from abstract.

P2, line 52: "pure zeolitic " >>> "untreated zeolitic" or "unmodified zeolitic"

P2, line 70: Is Aladdin a chemical company?

P2, line 80: Is the particle size average?

P2, lines 85 and 89: "for three times" >>> "three times"

P3, line 108: "were mixed with" >>> "was mixed with"

P3, line 145, "were also examined" >>> "was also examined"

P4. line 148: "[37, 38]. While" >>>  "[37, 38], while"

P5, line 159: "200mg/L of" Could you modify this sentence, because English sentence doesn't start with numerical value?

Reviewer 3 Report

All concerns have been carefully addressed. 

Author Response

Thanks again for your helpful suggestions.